# Genomic signatures of domestication in Old World camels

Robert Rodgers Fitak [1,15 ✉], Elmira Mohandesan[1,2], Jukka Corander[3,4,5], Adiya Yadamsuren[6,7],
Battsetseg Chuluunbat[8], Omer Abdelhadi[9], Abdul Raziq [10], Peter Nagy [11], Chris Walzer [12,13],
Bernard Faye [14] & Pamela Anna Burger [1,13 ✉]

Domestication begins with the selection of animals showing less fear of humans. In most domesticates, selection signals for tameness have been superimposed by intensive breeding for economical or other desirable traits. Old World camels, conversely, have maintained high genetic variation and lack secondary bottlenecks associated with breed development. By re-sequencing multiple genomes from dromedaries, Bactrian camels, and their endangered wild relatives, here we show that positive selection for candidate genes underlying traits collectively referred to as 'domestication syndrome' is consistent with neural crest deficiencies and altered thyroid hormone-based signaling. Comparing our results with other domestic species, we postulate that the core set of domestication genes is considerably smaller than the pan-domestication set – and overlapping genes are likely a result of chance and redundancy. These results, along with the extensive genomic resources provided, are an important contribution to understanding the evolutionary history of camels and the genomic features of their domestication.

[1] Institute of Population Genetics, Vetmeduni Vienna, Veterinärplatz 1, 1210 Vienna, Austria. [2] Department of Evolutionary Anthropology, University of Vienna, Althanstrasse 14, 1090 Vienna, Austria. [3] Wellcome Sanger Institute, Hinxton, UK. [4] Helsinki Institute for Information Technology, Department of Mathematics and Statistics, University of Helsinki, FIN-00014 Helsinki, Finland. [5] Department of Biostatistics, University of Oslo, N-0317 Oslo, Norway. [6] Institute of Remote Sensing and Digital Earth, Chinese Academy of Sciences, Jia No.20 North, DaTun road, ChaoYang District, Beijing, China. [7] Wild Camel Protection Foundation Mongolia. Jukov avenue, Bayanzurh District, Ulaanbaatar 13343, Mongolia. [8] Laboratory of Genetics, Institute of General and Experimental Biology, Mongolian Academy of Sciences, Peace avenue-54b, Bayarzurh District, Ulaanbaatar 210351, Mongolia. [9] University of Khartoum, Department for Meat Sciences, Khartoum, Sudan. [10] Camelait, Alain Farms for Livestock Production, Alain Dubai Road, Alain, United Arab Emirates. [11] Farm and Veterinary Department, Emirates Industry for Camel Milk and Products, PO Box 294236, Dubai, Umm Nahad, United Arab Emirates. [12] Wildlife Conservation Society, Wildlife Health Program, Bronx, NY, USA. [13] Research Institute of Wildlife Ecology, Vetmeduni Vienna, Savoyenstraße 1, 1160 Vienna, Austria. [14] CIRAD-ES, UMR 112, Campus International de Baillarguet, TA C/112A, 34398 Montpellier, France. [15] Present address: Department of Biology, Genomics and Bioinformatics Cluster, University of Central Florida, Orlando, FL 32816, USA. ✉email: rfitak9@gmail.com; Pamela.burger@vetmeduni.ac.at

The birth and ascent of human civilization can largely be attributed to the habituation and cultivation of wild plants and animals. By providing a more reliable stream of resources such as food and clothing, this process of domestication facilitated the shift from hunter-gatherer subsistence to that of agriculture. In animals, domestication likely occurred via multi-stage processes depending on the anthropophily of the wild ancestor (commensal pathway) and/or needs of humans (prey or directed pathways)[1]. Whether initiated by the wild animal ancestor or humans, intentional or not, the fundamental basis for domestication originated from a reduced fear of humans, i.e., tameness[2]. Thereafter, humans could continue the domestication process by breeding individuals harboring favorable traits through a process termed artificial selection. Domestication, however, is not limited to artificial selection, but also includes the relaxation of natural selection pressures such as predation and starvation, and the indirect, unintentional effects on traits correlated with captivity and those artificially selected[2]. In addition to tameness, the domestication of animals has led to a suite of morphological, physiological, and behavioral changes common to many species. These shared traits—including tameness, changes in coat color, modified reproductive cycles, altered hormone and neurotransmitter levels, and features of neotenization—are collectively referred to as the 'domestication syndrome' (DS)[3].

In general, two hypotheses have been proposed to govern the relationship between the development of DS and the underlying genes responsible. First, Crockford[4] suggested that the regulation of thyroid hormone concentrations during development may be linked to the neotenized phenotype of DS (thyroid hormone hypothesis; THH). The thyroid hormones triiodothyronine and its precursor tetraiodothyronine are produced during embryonic and fetal development, and also play key roles in postnatal and juvenile development[4,5]. The THH has been supported by research in domestic chickens, for example, where a fixed mutation in the thyroid stimulating hormone receptor gene has been extensively linked to the characteristic traits of DS[6].

The second hypothesis proposed by Wilkins et al.[3] predicts that DS is a consequence of mild deficits in neural crest cells during embryonic development; a product of artificial selection for behavior on standing genetic variation (neural crest cell hypothesis; NCCH). In horses, for example, selected genes were enriched for functions such as associative learning, abnormal synaptic transmissions, ear shape, and neural crest cell morphology, in addition to genes transcribed in brain regions containing neurons related to movement, learning and reward[7]. In cats, genomic regions under selection were associated with (i) neurotransmitters, responsible for serotonergic innervation of the brain, maintaining specific neuronal connections in the brain and fear conditioning, (ii) sensory development like hearing, vision and olfaction, and (iii) and neural crest cell survival[8]. Comparisons between genomes of village dogs and wolves also highlighted the role of neural crest cell migration, differentiation, and development in dog domestication[9]. Although evidence for both hypotheses exists, they are not necessarily mutually exclusive, and the relative contribution of each may vary along a continuum[5]. Furthermore, despite DS being generally shared among domesticated species, a universal set of underlying genetic initiators may not exist and each case of DS may arise from independent mechanisms. Extensive examination of the genes artificially selected by humans across a variety of species and conditions will help to advance the understanding of DS.

Old World camels offer a unique opportunity for studies of domestication because they have maintained relatively high levels of genetic variation, are largely multipurpose, and lack the secondary bottlenecks associated with specific breed development often characteristic of domestic species[10–13]. In essence, domestic Old World camels represent features of the "initial stages" of the domestication process, which were primarily focused on the selection for tameness and docility. Of the three extant species of Old World camels, two are domesticated (single-humped dromedaries, *Camelus dromedarius*, and two-humped Bactrian camels *Camelus bactrianus*) and one remains wild (two-humped wild camels *Camelus ferus*). The two-humped camels, *C. ferus* and *C. bactrianus*, shared a common ancestor ~1 million years before present (ybp)[14], whereas the common ancestor of all three Old World camelid species existed between 4.4 and 7.3 million ybp[14,15]. Domesticated camels are an essential resource, providing food, labor, commodities, and sport to millions of people. Furthermore, each species possesses a variety of adaptations to harsh desert conditions, including mechanisms to tolerate extreme temperatures, dehydration, and sandy terrain. Recent genomic studies of camels have identified patterns of selection consistent with the aforementioned adaptations[15,16], in addition to quantifying genetic variation and examining demographic history[15–18]. However, these studies are limited to analyses from a single genome of each species, thus biasing many inferences of selection and adaptation. For example, with a small sample size and closely related species, differences between sequences may not indicate fixation events but rather unobserved segregating polymorphisms; resulting in exaggerated estimates of the *Ka/Ks* ratio[19]. Furthermore, draft genomes are susceptible to errors in the estimated number of genes—thereby distorting conclusions of adaptation based upon orthologous genes between species (e.g., *Ka/Ks* ratio, gene expansion–contraction tests)[20].

In this study, we take a genomics approach to inferring both positive selection and demographic history of Old World camelids with an emphasis on genes potentially contributing to the DS phenotype. Considering that the direct wild ancestors of each domestic camelid (*C. dromedarius* and *C. bactrianus*) have been extinct for millennia, unlike in most other livestock, we inferred positive selection independently for each domesticated camelid using tests specific for the pattern of relationship between them and their wild counterpart (*C. ferus*). By re-sequencing multiple genomes from each species, we found evidence for positive selection on genes associated with both hypotheses of DS. These results, along with the extensive genomic resources made available, are an important contribution to understanding both the evolutionary history of camels and the underlying genomic features of their domestication.

## Results

**Whole genome re-sequencing.** We assembled a collection of 25 Old World camel samples from throughout their range to examine the patterns and distribution of genomic variation (Supplementary Fig. 1 and Supplementary Data 1). Our collection included representatives of all three extant species (Fig. 1a), *C. dromedarius* (*n* = 9), *C. bactrianus* (*n* = 7), and *C. ferus* (*n* = 9). In both domesticated varieties (*C. dromedarius* and *C. bactrianus*), we sampled across geographical regions in an attempt to minimize effects of genetic drift[8]. Using the Illumina HiSeq2000 system, we generated ~36 gb of sequence for each individual (Supplementary Fig. 2 and Supplementary Data 2). An average of 76% (SD 2%) of bases uniquely aligned to the *C. ferus* reference genome[16], resulting in mean genome coverage of 13.9×(SD 1.6×) per sample (Supplementary Data 2). When mapping reads to the *C. ferus* reference genome, no difference in genome coverage was observed among species (Kruskal–Wallis rank sum test, $\chi^2 = 3.58$, df = 2, *p* value = 0.167), but mean mapping quality varied significantly (Kruskal–Wallis rank sum test, $\chi^2 = 11.35$, df = 2, *p* value = 0.0034) (Table 1). To determine if the difference could be attributed to a larger divergence between

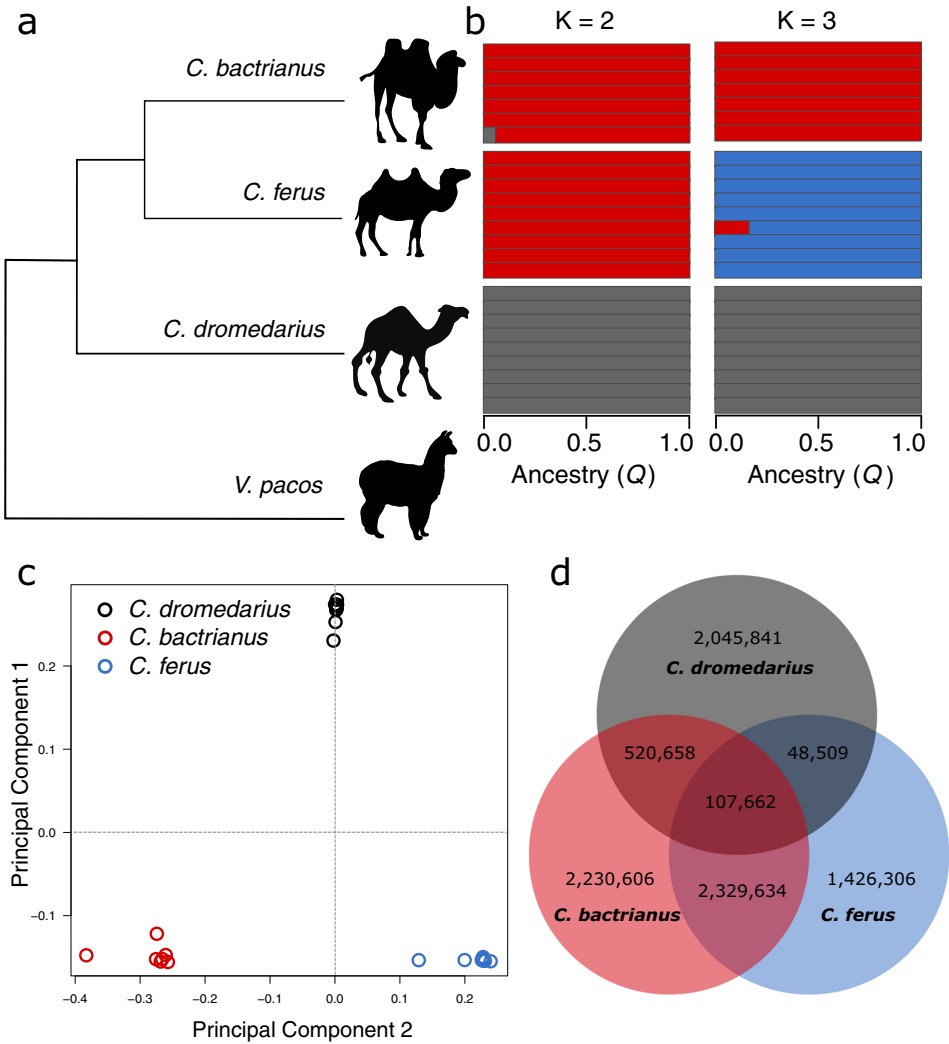

**Fig. 1 Distribution of genetic variation among Old World camels. a** Summary of the phylogenetic relationships among Old World camels using the alpaca (*Vicugna pacos*) as an outgroup. **b** Results from the population clustering analysis. Each horizontal bar corresponds with the proportion of ancestry in either two or three population clusters. **c** The first two principal components explain 23.3% and 10.1% of the SNP variation, respectively, and easily differentiate the three camel species. **d** Euler diagram showing the number of segregating SNPs unique to each species and shared between them. Note that the number of *C. bactrianus* individuals ($n = 7$) was less than the number of *C. ferus* and *C. dromedarius* individuals ($n = 9$). Approximately 2.1 million SNPs fixed between species are not included.

**Table 1 Summary of the mapping statistics and measures of genetic variation across the 25 camelid genomes sequenced.**

|  | C. ferus | C. bactrianus | C. dromedarius | C. dromedarius |
|---|---|---|---|---|
| *n* | 9 | 7 | 9 | 9 |
| Reference genome | C. ferus | C. ferus | C. ferus | C. dromedarius |
| Coverage | 14.6 (0.54) | 14.5 (0.44) | 12.7 (2.2) | 12.5 (2.2) |
| Mapping quality | 45.0 (1.82) | 44.3 (1.55) | 41.9 (1.13) | 42.3 (0.41) |
| $\theta$ ($\times 10^{-3}$) | 0.59 (0.38) | 0.85 (0.57) | 0.41 (0.34) | 0.41 (0.34) |
| $\pi$ ($\times 10^{-3}$) | 0.71 (0.48) | 0.88 (0.58) | 0.52 (0.44) | 0.52 (0.45) |
| Tajima's D | 0.75 (1.18) | 0.30 (1.22) | 0.95 (1.03) | 1.21 (0.89) |

The mean and standard deviation (within parentheses) are shown for each calculation using ~190,000 nonoverlapping 10-kb windows.

*C. dromedarius* and the other *Camelus* species[15], we repeated the mapping of reads from dromedaries to the *C. dromedarius* reference genome[18] (Table 1). The intraspecific mapping of dromedary reads resulted in a similar mean coverage (12.7× and 12.5× when mapped to *C. ferus* and *C. dromedarius* references, respectively) and no difference in mapping quality (Wilcoxon rank sum test, $V = 30$, $p$ value = 0.43). Mapping quality continued to vary among species (Kruskal–Wallis rank sum test, $\chi^2 = 14.33$, df = 2, $p$ value = 0.001) when including the intraspecific dromedary alignments. These results indicated that the relative consequences of mapping our dromedary sequences to either the *C. ferus* or *C. dromedarius* reference genomes were of

little concern. Previous studies of aligned camelid genomes found a high degree of synteny, including >90% coverage of *C. dromedarius* when aligned to either *C. ferus* or *C. bactrianus*, and concluded a majority of divergence can be attributed to single base mutations and small rearrangements[15,18]. As a result, subsequent analyses and comparisons across species were performed using dromedary reads mapped to the *C. ferus* reference, although for completeness, we report results from alignments to both species when calculating parameters within dromedaries.

**Comparison of genetic variation.** We identified single nucleotide polymorphisms (SNPs) after a series of recalibrating procedures that are known to improve variant detection[21]. The ratio of transitions to transversions (Ts/Tv) across all SNPs, a metric commonly used to assess variant quality, was 2.54—suggesting a high-quality set of variants[22]. Using our conservative approach, we identified ~10.8 million SNPs, of which 8.71 million (80.5%) were polymorphic within species and the remaining were fixed differences between species (monomorphic within a species, but differed between species). The illustration in Fig. 1d shows the number of polymorphic SNPs shared among species. The most segregating SNPs were observed in *C. bactrianus* (5.2 million), much higher than observed in *C. ferus* (3.9 million) and *C. dromedarius* (2.7 million) and despite sampling fewer individuals ($n = 7$ for *C. bactrianus* compared with $n = 9$ for the other two species). Of all the SNPs identified, only 107,662 (1.0%) were segregating within all three species. A large number of SNPs (~2.3 million) were shared between *C. bactrianus* and *C. ferus*, whereas much less were shared between *C. dromedarius* and *C. ferus* (48,509) than between *C. dromedarius* and *C. bactrianus* (520,658). Within *C. dromedarius* sequences aligned to the intraspecific reference, both Ts/Tv (2.49) and the number of SNPs (2,818,163) were comparable with the interspecific alignment (Table 1). The large number of SNPs shared between the domesticated species is consistent with known introgression events and has been observed in other genomic studies of camels[23]. Anthropogenic hybridization between the domesticated dromedary and Bactrian camel, especially in central Asia, is a widely practiced tradition of cross-breeding aimed at improving milk production (F1 backcrossed with dromedary), wool and meat yield, cold resistance (F1 backcrossed with Bactrian camel), or for camel wrestling[24,25].

The various summary measures of genetic variation across populations are provided in Table 1, along with individual level heterozygosity in Fig. 2a. The mean population mutation rate θ ($0.41 \times 10^{-3}$) and nucleotide diversity π ($0.52 \times 10^{-3}$) were reduced in *C. dromedarius* relative to the other species and greatest in *C. bactrianus* ($0.85 \times 10^{-3}$ and $0.88 \times 10^{-3}$, respectively). These patterns of genetic variation and heterozygosity are consistent with the relative differences observed between single genomes of the same species[15–18]. Interestingly, Ming et al.[23] reported similar patterns of genetic variation in *C. bactrianus* ($\pi = 0.95 \times 10^{-3} - 1.1 \times 10^{-3}$) and *C. ferus* ($\pi = 0.88 \times 10^{-3}$ compared with $0.71 \times 10^{-3}$ in this study), but nearly threefold higher levels in *C. dromedarius* ($\pi = 1.5 \times 10^{-3}$). Although this finding by Ming et al. conflicts with our results and that of the previously mentioned studies[15–18], the authors attributed this to the small sample size ($n = 4$) and being sampled from Iran where hybridization with *C. bactrianus* is commonplace.

**Demographic reconstruction.** We inferred historical changes in effective population size ($N_e$) using the pairwise sequentially Markovian coalescent model (PSMC)[26] (Fig. 2b). In dromedaries, the patterns of demographic history were nearly identical when using either *C. dromedarius* or *C. ferus* as the reference genome

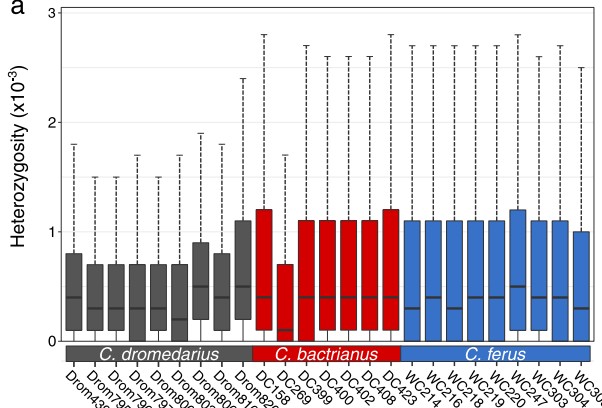

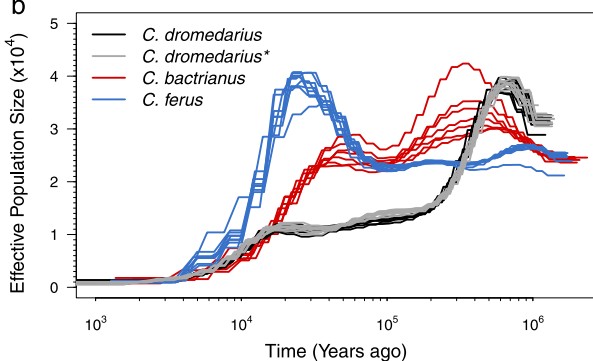

**Fig. 2 Heterozygosity and historical effective population sizes of the three extant camelid species. a** Boxplots of individual heterozygosity inferred from approximately 190,000 nonoverlapping 10-kb windows across the genome and colored by species. The boxplots' elements include the median (center line), first through third quartiles (colored box), and 1.5× interquartile range (whiskers). **b** The demographic history of each individual camel colored by species (see legend). All results were scaled using a generation time of five years and a mutation rate of $1.1 \times 10^{-8}$ (changes · site$^{-1}$ · generation$^{-1}$). The asterisk indicates that *C. dromedarius* analyses were repeated with mapping to the *C. dromedarius* reference.

sequence. Also, historical $N_e$ was remarkably similar across individual dromedaries and consistent with previous analyses from single genomes[15,18]. It appeared that dromedaries suffered a large bottleneck beginning around 700,000 ybp that reduced $N_e$ from nearly 40,000 to 15,000 by 200,000 ybp. The dromedary population further collapsed during and after the last glacial maximum (16,000–26,000 ybp), a finding shared with previous dromedary genomes and Northern Hemisphere mammalian megafauna[15,18,27].

Both *C. bactrianus* and *C. ferus* shared the same pattern of historical $N_e$ (~25,000) until 1 million ybp, matching the estimated divergence time (1.1 million ybp) reported between these species from mitogenomic sequences[14] and further supporting differentiation between these species prior to domestication by humans. Within *C. bactrianus*, $N_e$ peaked between 25,000 and 40,000 individuals ~400,000 ybp, and has since suffered a long-term decline over the last 50,000 years. The wild camel, on the other hand, experienced a large expansion between 50,000 and 20,000 ybp, but dramatically declined immediately thereafter; also consistent with potential effects of the last glacial maximum as observed in dromedaries. Although we interpreted the PSMC results assuming the demographic history represented changes in $N_e$ of a single population, these patterns can be confounded by past changes in gene flow or population structure, such as potential hybridization between

wild camels and the wild ancestor of extant domestic Bactrian camels. However, given that the population expansion and contraction timeline fits generally well with the previously estimated divergence time and the known climatic changes, the main PSMC based conclusions should be relatively robust.

**Population clustering**. We employed two methods to explore the global, or genome-wide average, ancestry of Old World camels using unlinked SNPs. First, we observed the clustering of individuals using principal components analysis (Fig. 1c). Coinciding with phylogenetic predictions (Fig. 1a), the largest component of variation (23.3%) separated one-humped *C. dromedarius* from its two-humped congeners, and the second component (10.1%) separated *C. ferus* from *C. bactrianus* (Fig. 1c). The remaining components accounted for increasingly less variation (<6%) and tended to separate single individuals (Supplementary Fig. 4). The analysis does suggest a slight potential for introgression between species, notably between a single *C. ferus* individual and *C. bactrianus*. The second method employed a Bayesian model-based approach[28] to clustering individuals and indicated high support for both two and three genetic clusters that corresponded with each species (Fig. 1b). Similar to the principal components analysis, the results indicated that a small amount (6.0%) of *C. dromedarius* ancestry is present in a *C. bactrianus* individual, and larger amount of *C. bactrianus* ancestry in a *C. ferus* individual (16.6%). These values are markedly similar to values reported by Ming et al.[23], where *C. dromedarius* ancestry in *C. bactrianus* ranged between 1 and 10%, and *C. bactrianus* ancestry in three *C. ferus* individuals ranged between 7 and 15%. Introgression from the domestic species into both Mongolian and Chinese populations of the Critically Endangered wild camel has been reported elsewhere using mitochondrial DNA[29], microsatellites[30] and the Y chromosome[31], potentially jeopardizing the wild camel's genomic integrity and evolutionary independence (~1.1 million years)[14].

**Positive selection in dromedaries**. We identified candidate genes under positive selection in dromedaries using a combination of tests based on the ratios of polymorphism to divergence within genes (homogeneity and Hudson–Kreitman–Aguadé tests[32]) and among genomic windows (low nucleotide diversity [$\pi$] and high divergence [$D_{XY}$]). For the former, we were able to examine 10,297 (57.5%) genes with at least one fixed variant and 84 (0.82%) genes passed our criteria for positive selection (Supplementary Data 3 and 4). Using the latter approach, we identified 17 100-kb genomic windows containing both the lowest 0.5 percentile of $\pi$ and highest 99.5 percentile of $D_{XY}$. These windows contained a total of 23 protein-coding genes (Supplementary Data 5 and 6). For both sets of putative positively selected genes, we found no significant enrichment of any category of gene ontology (GO) terms or KEGG pathways after correction for multiple comparisons (Supplementary Data 4 and 6).

Among the 107 genes linked to recent, positive selection, a variety of functions and processes were represented, including numerous neurological and endocrine functions as predicted by the NCCH and THH during domestication. At least *TUBGCP6*, *SYNE1*, *BPTF*, *KIDINS220*, *MYO5A*, *VPS13B*, *TBC1D24*) are known to contain mutations causing various neuropathies (Supplementary Data 3 and 5). These neuropathies often manifest as features characteristic of DS such as microcephaly, facial dysmorphism, and intellectual disability (e.g., Cohen syndrome caused by a mutation in *VPS13B*). The genes *VPS13B* and *BPTF*, in particular, have also been reported as linked to domestication from genomic scans of chicken[33] and dogs[9], respectively (Fig. 3a). Four genes were previously identified as under positive selection

in both *C. dromedarius* and *C. bactrianus* (*CENPF*, *CYSLTR2*, *HIVEP1*) or just in *C. dromedarius* (*CCDC40*)[15], suggesting either a convergent consequence of domestication, or a more general role in camel adaptation. The latter is more likely, considering that *CENPF*, *CYSLTR2*, and *CCDC40* are each related to ciliopathies and/or respiratory diseases such as asthma—indicating a possible adaptation to the respiratory challenge posed by dust in highly arid environments[15].

Several other genes (*SSH2*, *CABIN1*, *NOS1*, *NEO1*, *INSC*, *EXOC3*) were also known to have important functional and/or developmental roles in the neural system. For example, *SSH2*, which is also linked with domestication in dogs[9], is a phosphatase critical for neurite extension[34]. In the mammalian brain, the gene *NOS1* is an important neurotransmitter[35] and *EXOC3* is active in neurotransmitter release[36]. Both genes *CABIN1* and *NEO1* are critical during development for the proper migration of neural crest cells[37,38]—a key prediction of the NCCH[3]. Of special interest is the gene *ATRN*, whose pleiotropic effects include pigmentation phenotypes (e.g., the mahogany phenotype in mice)[39] and a crucial role in the proper myelination of the central nervous system[40]. *ATRN* has also been reported as under positive selection during yak domestication[41] and during long-term experimental selection for tameness in foxes[42].

Among the candidate positively selected genes, several potentially associated with the THH were identified (*PDPK1*, *PLCD3*, *CCNF*, *GFRA4*). Both *PDPK1* and *PLCD3* are components of the thyroid hormone signaling pathway (KEGG pathway ko04919), and *PDPK1* expression is known to be increased in follicular cell thyroid carcinoma in dogs[43]. The gene *CCNF*, which in dromedaries contained two, fixed non-synonymous substitutions, is known to be regulated by thyroid hormone during development[44]. In addition, *CCNF* participates in the ubiquitination and targeting of certain proteins for degradation and has been linked to neuronal degeneration disorders such as congenital amyotrophic lateral sclerosis[45]. The *GFRA4* is not only important from a neurological perspective as it binds neurotrophic factors in the *GDNF/RET* signaling pathway, but also the expression and splicing of *GFRA4* has been linked to endocrine cell development, including the thyroid where its expression is localized in adult humans[46].

**Positive selection in domestic Bactrian camels**. In *C. bactrianus*, it was possible to perform the homogeneity test above in only 90 genes because the close relationship with *C. ferus* resulted in very few fixed differences within genes thus reducing the power of the test. Of these 90 genes, none showed evidence of positive selection in *C. bactrianus*. To mitigate this issue, we subsequently tested for excessive allele frequency divergence in *C. bactrianus* relative to *C. ferus* and using *C. dromedarius* as an outgroup. This test, known as the population branch statistic (PBS)[47], produced 39 windows that passed our criteria for positive selection and overlapped ten protein-coding genes (Supplementary Data 7). In addition, as performed for dromedaries, we identified two 100-kb windows with excess $D_{XY}$ and a dearth of $\pi$. No protein-coding genes were found in these regions. Although the ten putative positively selected genes were not enriched for any specific GO functions (Supplementary Data 8) or KEGG pathways, several genes were promising candidates for associations with camel domestication. For example, the histone demethylase *KDM1A* regulates global DNA methylation and the expression of many genes via chromatin remodeling[48]. Like several of the genes found in dromedaries, defects in *KDM1A* cause craniofacial disorders and psychomotor retardation[49]—again, signature features of DS. Furthermore, *KDM1A* is required for pituitary organogenesis[50], and stress hormone activity

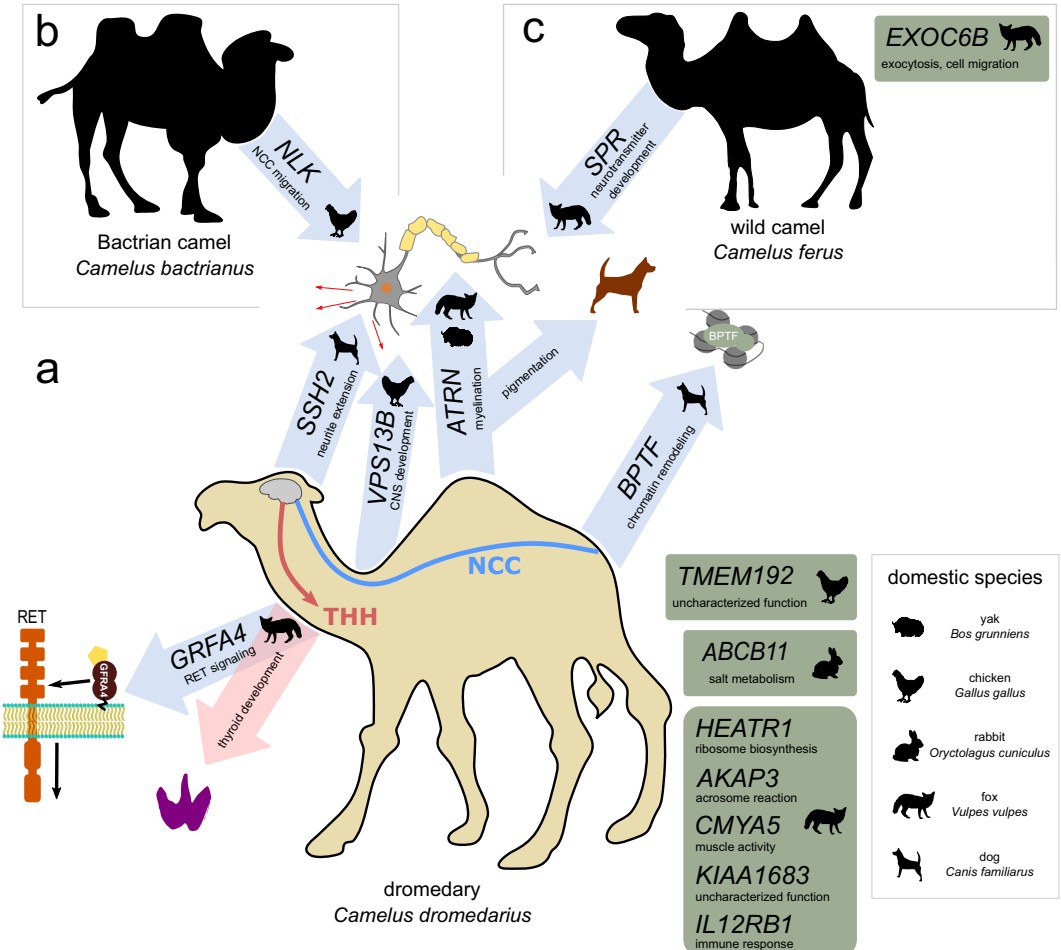

**Fig. 3 Genes under selection in Old World camels that overlap candidate domestication genes in other domestic species.** The genes under positive selection in **a** *Camelus dromedarius*, **b** *C. bactrianus*, and **c** *C. ferus* were of functions that could be associated with the neural crest cell (NCC; blue) and thyroid hormone (pink) hypotheses of domestication syndrome, or of other, unknown relationship (green). Genes are shown with an icon of the corresponding domestic species where positive selection has also been reported. Arrows are directed toward a summary icon of known functions for that gene.

regulated by the hypothalamic–pituitary–adrenal axis is also tightly correlated with both thyroid hormone activity[51] and tameness in foxes[52]. The genes *LUZP1* and *NLK* both function in neural development. Mouse knockouts of *LUZP1* develop neural tube closure defects in the embryonic brain[53], and *NLK*, also selected in domestic chickens[33] (Fig. 3b), acts as part of the noncanonical *Wnt*/Ca$^{2+}$ pathway to inhibit canonical *Wnt*/ß-catenin and control the migration of neural crest cells[54,55].

**Relaxed selection relative to wild two-humped camels.** In wild camels we undertook a different approach to identify genomic regions that signify relaxed selection in the domestic species. These windows had an excessive π log-ratio $(\ln \pi_{domestic} - \ln \pi_{wild})$ in both domestic species (99.5 percentile) and a substantially negative measure of Tajima's D (D ≤ −2) in *C. ferus*. Three 100-kb windows passed these conservative criteria and one of the windows contained two protein-coding genes, *SPR* and *EXOC6B* (Fig. 4; Supplementary Data 9). A region containing both these genes has also been described as differentiating between tame and aggressive foxes[42] (Fig. 3c). In humans, deficiencies in *SPR* cause dystonia—uncontrollable muscular contractions—, psychomotor retardation, and progressive neurologic deterioration[56]. The gene *EXOC6B* is part of the exocyst complex, which is critical for cellular trafficking, and mutations in *EXOC6B*

have been associated with intellectual disability, language delay, hyperactivity, ear malformations, and craniofacial abnormalities in humans[57].

## Discussion

Old World camels represent an interesting example in understanding the genetic impacts of domestication. Camel breeders have aimed to retain a high degree of phenotypic diversity in their herds and generally avoided selection at the level of individual animals, with the exception of traits for tameness and tolerance of humans[10]. Without the secondary bottlenecks associated with specific breed formation, camels thus represent an initial stage in the domestication process. Our genomic scans for selection in two domesticated camel species identified candidate genes whose functions were consistent with many features of DS (e.g., neotenization, intellectual disability, neuropathies). More specifically, we found evidence of selection in genes that were associated with both the NCCH and THH, and shown to be under selection during the domestication of other species (i.e., chicken, yak, dog, fox, rabbit). These results prompted two important conclusions. First, the results supported that the NCCH and THH need not be mutually exclusive—the pathways are not completely independent and their relative contributions can vary across domestication events in space and time[5]. Second, the results supported that

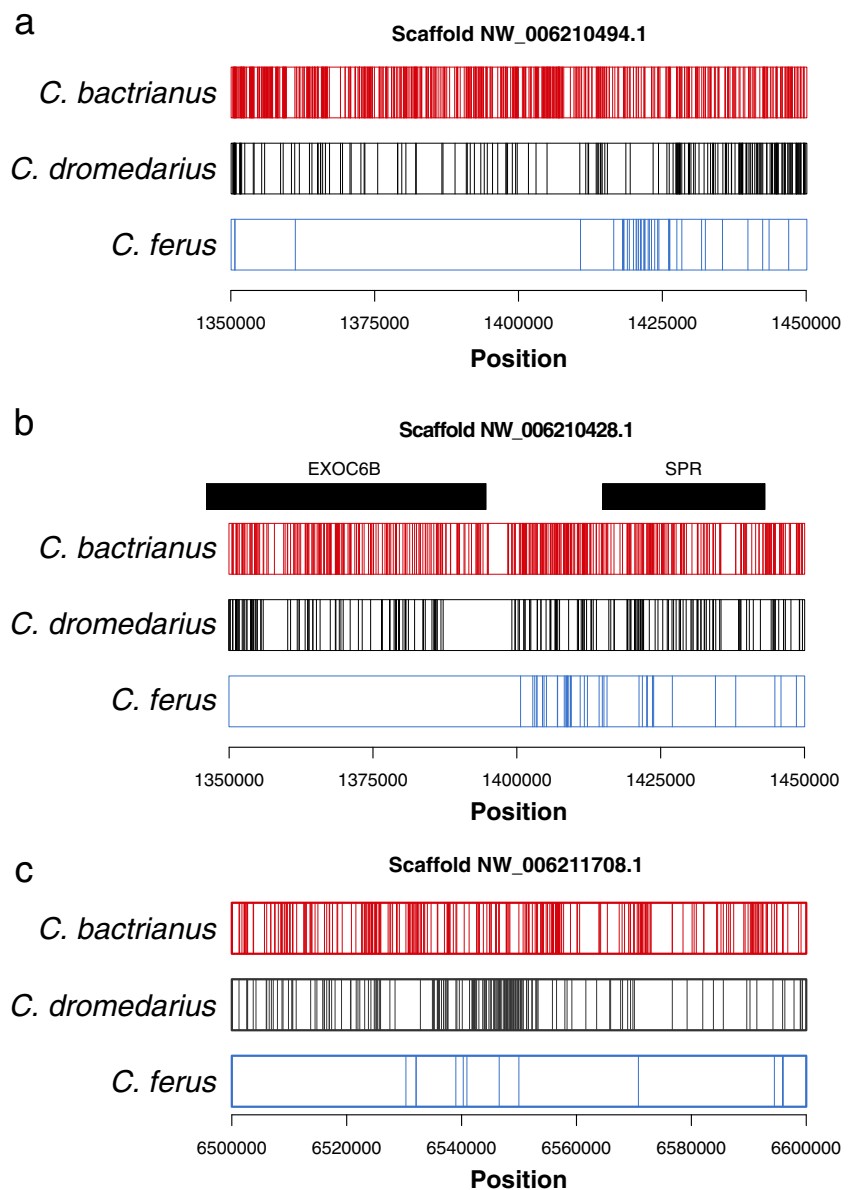

**Fig. 4 Positive selection in wild camels.** The three 100-kb windows in *Camelus ferus* (**a**–**c**) found to contain evidence of positive selection. The location of variants within each species are shown as vertical lines, and overlapping genes as black bars.

a shared set of domestication genes between camels does not exist, even across independent domestication processes of two evolutionary close species (divergence time ~4.4–7.3 mya[14,15]). It is possible that because the direct wild ancestors of each domestic camel are extinct, our tests for selection may recover positive selection occurring prior to domestication. Although we cannot completely exclude this possibility, we chose tests for selection that detect more recent events based upon comparisons of polymorphism to divergence, and we did identify genes associated with domestication in camels that overlapped those described in other domestic species. Future studies targeting the analysis of insertion/deletion (indel) polymorphisms may also be useful for identifying additional targets of selection. Indel polymorphisms were omitted from our analyses because their accurate genotyping remains quite challenging, especially without high coverage (≥60×[58]) and in non-model species with incomplete, or draft genome assemblies. With regards to domestication selection, a specific, universal set of domestication genes may not exist, but we do speculate that there may be a 'core' set of genes shared

across multiple domestication processes. This core set of genes, however, is small compared with the set of 'pan'-domestication genes (the sum of all genes selected across all domestication events). Nonetheless, the wealth of genomic data across domesticated animals suggests that future meta-analyses are warranted to determine the components of this core and pan-domestication genome.

Patterns of demographic history across all three species demonstrated widespread population declines during the late Pleistocene. Although the exact reason for these declines is unclear, it is consistent with declines and extinctions in other megafauna as a result of either climatic changes or human persecution. Dromedaries, in particular, have experienced a long-term, exceptional decline (possibly to as few as six maternal lineages[13]) reducing their nucleotide diversity to nearly half that of the other species. Wild camels, despite having more genetic variation than dromedaries, remain one of the most critically endangered of all mammals and are at substantial risk of extinction resulting from their continued population decline.

Our study contributes a large set of genomic resources for Old World camels. These resources, combined with the existing and ongoing development of other resources (e.g., chromosome-level assemblies, SNP chip), will aid in the prudent application of breeding and selection schemes to conserve and manage the genomic diversity of camels. In turn, these efforts will preserve the evolutionary potential of the wild species in addition to the promise and sustainability of domestic camels as a valuable livestock in arid environments.

## Methods

**Sample collection and sequencing**. We collected EDTA-preserved blood from 25 Old World camels including nine dromedaries (*C. dromedarius*), seven domestic Bactrian camels (*C. bactrianus*), and nine wild camels (*C. ferus*) (Supplementary Data 1 and Supplementary Fig. 1; see Ethics statement below). We included domesticated camels that represent a variety of geographic locations and/or 'breeds' according to information supplied by the camels' owners, although microsatellite evidence for both dromedaries and domestic Bactrian camels suggest little genetic differentiation among breeds[11–13]. In dromedaries, moderate differentiation exists between camels from northwest Africa (e.g., Canary Islands and Algeria), the Horn of Africa (e.g., Ethiopia, Somalia, Kenya), and their remaining range including northeast Africa, the Middle East, and Pakistan[11,13]. We extracted DNA using the Master PureTM DNA purification kit for blood (Epicentre version III) and generated a 500 bp paired-end library for each sample. We sequenced each library with a single lane of an Illumina HiSeq (Illumina, USA) according to standard protocols. Due to the sampling procedure, which included a wild endangered species, it was not possible to retrieve tissue samples that would facilitate expression studies and/or functional analyses. A follow-up project will consider this next analytical step.

**Read processing and alignments**. We trimmed the 3′ end of sequence reads to a minimum phred-scaled base quality score of 20 (probability of error <1.0%) and excluded trimmed reads <50 bp in length using POPOOLATION v1.2.2[59]. We aligned all processed reads to the *C. ferus* CB1 reference genome (Genbank accession: GCA_000311805.2) using BWA v0.6.2[60] with parameters '-n 0.01 -o 1 -e 12 -d 12 -l 32'. We removed duplicate reads and filtered alignments to only include reads that are properly paired and unambiguously mapped with a mapping quality score >20. We realigned reads around insertions/deletions and performed a base quality score recalibration using the Genome Analysis Toolkit (GATK) v3.1-1 following guidelines presented by Van der Auwera et al.[21]. As input into the base quality score recalibration step we generated a stringently filtered set of SNPs using the overlap of three different variant-calling algorithms (SAMTOOLS v1.1[61]; GATK HAPLOTYPECALLER v3.1-1[21]; ANGSD v0.563[62]). The overlapping SNPs were filtered to exclude those with a quality score (Q) < 20, depth of coverage (DP) > 750 (~30×/individual), quality by depth (QD) < 2.0, strand bias (FS) > 60.0, mapping quality (MQ) < 40.0, inbreeding coefficient < −0.8, mapping quality rank sum test (MQRankSum) < −12.5, and read position bias (ReadPosRankSum) < −8.0. Furthermore, we excluded SNPs if three or more were found within a 20-bp window, were within 10 bp of an insertion/deletion, or were found in an annotated repetitive region.

**Identification of sex chromosome-linked scaffolds**. We identified scaffolds from the reference genome that can putatively be assigned to the sex chromosomes (the reference genome was male) (Supplementary Fig. 3). This was a necessary step in order to remove variants from downstream analyses that require accurate estimates of allele frequencies assuming diploid samples (our samples consisted of both the homogametic and heterogametic sexes). We first aligned all scaffolds to the cattle X and Y assemblies (UMD3.1 and Btau4.6.1 assemblies, respectively) using LASTZ v1.02.00[63] with parameters '--step=1 --gapped --chain --inner=2000 --ydrop=3400 --gappedthresh=6000 --hspthresh=2200 --seed=12of19 --notransition'. For each scaffold with high-scoring alignments, we calculated the ratio of the scaffold coverage to the genome-wide mean coverage in each individual. To assign scaffolds to the X chromosome, we identified scaffolds whose coverage ratio in males was significantly less than the ratio in females using a Wilcoxon Rank Sum test ($p < 0.05$) and whose total alignment length was ≥20% of the total scaffold length. For the Y chromosome, we identified scaffolds whose coverage ratio did not differ significantly from 0.5 in males and was significantly less than 0.5 in females using a Wilcoxon Rank Sum test ($p < 0.05$) and whose total alignment length was ≥20% of the total scaffold length.

**Variant identification**. We generated another set of SNPs from the realigned and recalibrated alignment files using the GATK HAPLOTYPECALLER and filtering criteria as described above. We further excluded SNPs on scaffolds putatively assigned to the X and Y chromosome (see 'Identification of Sex Chromosomes' below), with a minimum allele count <2, missing a genotype in more than five individuals, with 4 > DP > 30 per genotype, and deviating from Hardy-Weinberg equilibrium ($p < 0.0001$) in VCFTOOLS v0.1.12b[64]. We used this set of SNPs as a training set to perform variant quality score recalibration in GATK, assigning a

probability of error to the training set of 0.1. This recalibration develops a Gaussian mixture model across the various annotations in the high-quality training dataset then applies the model to all variants in the initial dataset. After variant recalibration, we excluded all SNPs with VQSLOD score outside the range containing 95% of the SNPs in the training set. We hard-filtered any remaining variants missing genotypes in more than five individuals. Variants on the X and Y putative scaffolds were excluded from all analyses except for gene-based analyses described below (See Homogeneity and HKA tests below).

We assessed the quality of the final set of SNPs by calculating the ratio of transitions to transversions (Ti/Tv ratio) in VCFTOOLS. The Ti/Tv ratio is often used as a diagnostic parameter to examine the quality of SNP identification[22]. When substitution is random Ti/Tv = 0.5 because there are twice as many transversions possible as transitions. However, in humans the genome-wide Ti/Tv is ~2.0–2.2, so values much less than this are indicative of an excess of false-positive SNPs[22]. Estimates of genetic variation (i.e. π, θ, and heterozygosity) and Tajima's D were averaged across nonoverlapping 10-kb windows using VCFTOOLS and excluding SNPs on the putative X and Y scaffolds.

**Population clustering**. To infer population clustering, we used SNPRELATE 1.10.1[65] to calculate linkage disequilibrium between pairs on SNPs within a 1-Mb sliding window and randomly removed one locus from each pair with a correlation coefficient ($r^2$) >0.5. The resulting dataset contained 90,918 unlinked SNPs. We used the unlinked SNPs to examine the global ancestry of Old World camels using the principal components analysis method in SNPRELATE and a Bayesian model-based approach implemented in ADMIXTURE v1.23[28]. We restricted the Bayesian analysis to the maximum likelihood estimation of individual ancestry proportions (Q values) in three ancestral populations. Likelihood searches were terminated for each point estimate when the log likelihood increased by less than 0.0001 between iterations (parameters: -C 0.0001 -c 0.0001).

**Demographic inference**. We used PSMC (v0.6.4[26]) to examine the demographic history of the three camelid species. The PSMC model infers the historical effective population size ($N_e$) from a single diploid genome by examining the distribution of coalescent rates across the genome. Because the coalescent rates across the genome are dependent upon the density of polymorphic sites, we employed a strict set of conditions as described previously by our group[18]. Briefly, for each individual we first constructed a genome sequence by applying the individual-specific alleles from our final set of SNPs to the *C. ferus* reference genome. Furthermore, we masked all repetitive regions and putative X and Y contigs from the analysis. We ran PSMC for a total of 25 iterations using the parameters '-t15 -r5 -p "4 + 25 × 2 + 4 + 6"' and verified that ~10 recombination events occurred in the final set of intervals spanned by each parameter[26]. We ran 100 bootstrap replicates to assess the variance in the final inference of $N_e$. For *C. dromedarius*, we repeated the PSMC analysis as described above using the intraspecific reference. We scaled the final results using a generation time of five years and mutation rate of $1.1 \times 10^{-8}$.

**Signals of positive selection**. We employed multiple approaches to identify candidate genes under positive selection in domesticated camel species. First, we used a gene-based approach that combined the homogeneity test and the HKA test[32]. The homogeneity test examines the intraspecific (polymorphism) and interspecific (divergence) genetic diversity, which are expected to be correlated under neutral evolution. Under positive selection, the amount of polymorphism is expected to be reduced along one branch. To perform the test in dromedaries, we calculated four values for each of the 17,912 protein-coding genes (longest isoform per gene) annotated in the camel genome:

(A) Number of polymorphic sites in the dromedary samples.
(B) Number of polymorphic sites in the wild camel samples.
(C) Number of fixed differences between dromedaries and both wild camels and the alpaca genome sequence.
(D) Number of fixed differences between wild camels and both dromedaries and the alpaca genome sequence.

We then tested the null hypothesis that $\frac{A}{C} = \frac{B}{D} \equiv \frac{A}{B} = \frac{C}{D}$ using a Fisher exact test for a 2 × 2 contingency table. We omitted any genes with either A or C < 1. Alpaca alleles were identified by mapping all short-insert, paired-end sequencing reads from the alpaca genome assembly[15] (BioProject accession PRJNA233565) to the camel reference genome as described above. Then, for each camel SNP location, we selected the most common allele (minimum depth of two) from the aligned alpaca reads. If multiple bases occurred at equal frequency, one was randomly selected. Next, we performed the HKA test by comparing ratio $\frac{A}{C}$ for each gene to the ratio $\frac{A}{C}$ summed across all genes analyzed ($\frac{A}{C}[gene] = \frac{A}{C}[genome]$) using a Fisher exact test. For the final set of putative positively selected genes, we retained those with a homogeneity test $P < 0.05$ and with a significant HKA test score ($P < 0.05$) only in the dromedary population. As suggested by Liu et al.[32] the P values obtained from these tests can be misinterpreted since accurate P values can be only be obtained from simulations, but can be informative when combined with other ranking criteria. In conjunction with the recommendation by Liu et al. we emphasize that these genes are in ranked order of priority, or evidence, rather than of statistically

significant effect. The above procedure was repeated for domestic Bactrian camels using the wild camels again for comparison.

The second approach to identify putative positively selected genes utilized a window-based approach. We calculated nucleotide diversity ($\pi$) within and divergence ($D_{XY}$) between camelid species across 100-kb sliding windows with a step size of 50 kb using the popgenWindows.py script (https://github.com/simonhmartin/genomics_general). Only windows with at least ten polymorphic sites were included. Next, we defined candidate positively selected regions in dromedaries and domestic Bactrian camels as windows in both the lowest 0.5 percentile in $\pi$ within species and the highest 99.5 percentile in $D_{XY}$ relative to the wild camel population. Within domestic Bactrian camels only, we calculated the PBS[47]. The PBS is a powerful method to detect both complete and incomplete selective sweeps over relatively short divergence times assuming two populations and an outgroup[47]—making this test applicable for domestic Bactrian camels. Using the windows defined above, we calculated Reynold's $F_{ST}$ for the three population pairs and converted them to divergence times scaled by $N_E$ using the Cavalli-Sforza transformation $T = -\log(1 - F_{ST})$ [47]. The PBS was subsequently obtained from $PBS = \frac{T_1 + T_2 - T_3}{2}$, where $T_1$ is from the domestic Bactrian vs wild camel comparison, $T_2$ is from the domestic Bactrian vs dromedary comparison, and $T_3$ is from the wild camel vs. dromedary comparison. Windows in the top 99.5 percentile of PBS values were retained as positively selected. In *C. ferus*, we calculated the $\pi$ log-ratio with both domestic species ($\ln \pi_{domestic} - \ln \pi_{wild}$)[41] to identity windows with a relatively low level of polymorphism. As above, windows were 100 kb in length with a step size of 50 kb, and only windows with at least ten polymorphic sites across all species were retained. If a window contained no heterozygous sites within a species ($\pi = 0$), then a value less than the minimum across polymorphic windows was used ($\pi = 10^{-5}$) to avoid logarithmic errors. In addition, in *C. ferus* we calculated Tajima's D in each window. As a conservative estimate of regions undergoing positive selection in *C. ferus* and/or relaxed selection in the domestic species, we retained a final set of windows with an excessive $\pi$ log-ratio in both domestic species (99.5 percentile) and a substantially negative measure of Tajima's D ($D \leq -2$) in *C. ferus*. For all window-based analyses, protein-coding genes that overlap these windows were identified.

**Functional enrichment**. We assigned GO terms to all annotated protein-coding genes in the *C. ferus* reference genome using BLAST2GO v3.0.8[66]. BLASTP v2.2.30 (http://ncbi.nlm.nih.gov/blast) searches were conducted against metazoan protein sequences from the 'nr' database with an *e* value cut-off of $10^{-3}$ and retaining only the top 20 hits. We tested for functional enrichment of GO terms using topGO v2.28.0[67] with a classic Fisher exact test and minimum annotation count of five for each GO term across the full annotation set. We also tested each set of putative positively selected genes for overrepresentation of KEGG pathways using WEB-GESTALT[68]. In WEBGESTALT, we used the *Bos taurus* KEGG annotations of protein-coding genes as a reference set and otherwise default parameters. In all analyses we corrected for multiple testing using a false discovery rate <0.05. Additional functional information for each protein-coding can be found in Supplementary Data 10.

**Statistics and reproducibility**. Summary statistics and tests were calculated using R v3.6.2. The tests included the nonparametric Wilcoxon rank sum and Kruskal–Wallis rank sum tests for comparing mapping results within and between *C. dromedarius* ($n = 9$), *C. bactrianus* ($n = 7$), and *C. ferus* ($n = 9$). The full test results are reported in the Results section. Contingency table ($2 \times 2$) testing (homogeneity and HKA tests) was also performed in R using the Fisher exact test, and although are reported in Supplementary Data 3, *P* values were only used for relative prioritization rather than assessing statistical significance.

**Ethics statement**. The blood samples for each camelid species were retrieved during routine veterinary procedure, micro-chipping, or radio-collaring of Mongolian wild camels. All domestic and wild Bactrian camel samples were collected within the framework of the legal requirements of both Austria and Mongolia. Micro-chipping of wild camels from the breeding center of the Wild Camel Protection Foundation was performed with the request and consent of the foundation (John Hare, personal communication). Capture and collaring of wild camels within the Great Gobi Strictly Protected Area "A" was conducted within a cooperation agreement between the International Takhi Group and the Mongolian Ministry of Nature, Environment and Tourism signed on 15.02.2001 and renewed on 27.01.2011.

**Reporting summary**. Further information on research design is available in the Nature Research Reporting Summary linked to this article.

## Data availability
Raw and mapped sequencing data for the 25 genomes in the study are available in GenBank (BioProject accession PRJNA276064), and additional data have been deposited in Dryad (https://doi.org/10.5061/dryad.prr4xgxj2).

## Code availability
Computer code and scripts for the various analyses are available at GitHub (https://github.com/rfitak/Camel_Genomics).

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

## Acknowledgements

We thank all camel owners for dedicating aliquots of commensally collected samples for research purpose, especially J. Hare and K. Rae from the Wild Camel Protection Foundation, G. Gassner (Austria), J. Burgsteiner (Austria), A. and J. Perret (Kenya) and R. Saleh (Syria) for facilitating sample collection and for their continuous support. We thank the CSC—IT Center for Science, Finland, for generous computational resources. PAB acknowledges support by the Austrian Science Fund (FWF) project grants P24706-B25 and P29623-B25.

## Author contributions

R.R.F wrote the paper and performed bioinformatic analyses. E.M. extracted DNA, performed bioinformatic analyses, and revised the paper. J.C. provided the extensive computational resources necessary for the completion of the project and revised the paper. A.Y, B.C., O.A., A.R., P.N., C.W., and B.F. facilitated commensal sample collection and revised the paper. P.A.B. managed the project, carried out initial raw data analysis and wrote parts of the paper.

## Competing interests

The authors declare no competing interests.
