## [Peer Review File · Communications Biology]

Editorial Note: Parts of this Peer Review File have been redacted to remove confidential information.

Reviewers' comments:

Reviewer #1 (Remarks to the Author):

The authors analyze genetic diversity within and among three camel species to infer aspects of their population history and to identify genes which may have been targeted by selection, particularly during the domestication of two of the species. The results are presented in the context of the domestication syndrome and various proposed hypothesis in terms of gene pathways whose alteration lead to domestic phenotypes. This makes the findings likely to be of interest to a broad audience. As such, the manuscript would benefit from inclusion of additional relevant details about the history and divergence of the studied species. For example, the divergence between bactrianus and ferus (~1 MYA) should be made clear to highlight for the reader that ferus is not a proxy for the 'pre-domesticated ancestor' of bactrianus and to aiding in the interpretation of the used selection statistics. Similarly, the phylogenetic relationship to dromedaries at ~5-8 MYA should be emphasized in the text to aid the reader. If I understand correctly, this makes dromedaries a separate domestication from a genetically quite distinct and diverged ancestor vs the two-hump camels. This greatly complicates the comparisons among the two domesticated species. Because of these differences, the tests for selection, and the signals and time scales that might be detected, are quite different.

Further, because of the time scales, it is a concern that the tests may identify effects of natural selection along the lineages (millions of years) instead of recent effects associated with domestication. The logic excluding this possible confounding should more clearly be described.

Since there are multiple genomes available, the authors should consider using MSMC (Schiffels 2014) to both obtain population size profiles for more recent times and to better define the dynamics of population separation between the species.

I believe that alpaca is also considered to be domesticated, the author's may which to comment on what impact the use of a domesticated samples as an outgroup may have.

Reviewer #2 (Remarks to the Author):

The presented data do not fully support the main conclusions of camel domestication because they created a big gap in sampling, by the way the paper has worthwhile information that is useful for the signature of selection in domestication specially in old world camels , and there are essential points missing in the manuscript which need to be addressed in detail.

Line 137: The sequencing depth is low, while as reported the WGS data with a medium degree of depth setting (more than 15×) can be considered as one of the recommendation for SNV calling," Kishikawa, Toshihiro, et al. "Empirical evaluation of variant calling accuracy using ultra-deep whole-genome sequencing data." Scientific reports 9.1 (2019): 1784."

Line 137: the authors should explain more about the reason for using camelus ferus as a reference genome in alignment.

Line 163: the ratio ts/tv appears to be high (2.54) that usually it happens in exome studies or small genomes with high GC for example turkey genomes, the authors should explain about this.

Line 165: the author would be defining fixed differences (SNP or variants) to clarify for public readers.

Line 165-170: when you used the c. Ferus as reference genome, we expected the numbers of variants in the dromedary vs ferus set(2722670SNP) should be more than Bactrianus vs ferus (5188560SNP) and frus vs ferus sets(3912111 SNP), but it was vice versa. The author must explain why it happens.

In addition recheck the number of total variant (10.8 million) that seems to be 11.8.
Also, it would be worthwhile to add the result of InDels and compare between species.
Line 355: the distribution of sampling for decisions about genomic signature of domestication is not adequate because the author create long gaps of the world sampling (north Africa, IRAN, and other central Asia for dromedary and china, Russia, Iran and etc. for Bactrian) that are important gene pool for camels, in my suggestion if it is difficult to sampling of these aria, the authors could get the deposited data through NCBI.
Line 392: what is the reason of using the cattle as reference for alignment for X and Y chromosome, the author would be explaining about this.
In the end of comments, the authors compare the positive selected genes with other species such as fox, dogs but it would be useful to compare with camels that reported by Wu et al., 2014.

Reviewer #3 (Remarks to the Author):

The paper is very well-written, and provides novel insights into the domestication of camels. This has implications for genetic pathways involved in domestication in other species as well, which is certainly important for people in the field. The methods seem appropriate and technically sound, and are very well-documented. The results are interpreted appropriately, and context is provided for a general understanding of each part. In principle, only few things should be considered:

1) Major point: A study on 128 whole genome sequences of Asian camels was published on bioRxiv recently (doi:10.1101/656231v1). It would be good to discuss their findings regarding demographic history and admixture (methods like PCA or PSMC were also applied there), since there is an overlap in scope and results. This does not imply the results were identical or mutually exclusive, but seem rather complementary, and it would be important to be discussed here. Despite being available only on bioRxiv, it is common in population genetics to consider such manuscripts, and I recommend doing so.

2) Another major point: I cannot follow the reason for applying three different selection tests on the three branches. It would make sense to at least provide the results for all types of tests for all three branches as Supplementary, and provide a rationale for focusing on specific tests for each population. At least for *C. bactrianus* and *C. ferus*, there should be the same power for the same tests, for example using the PBS statistic.

3) It should be noted that the applied methods (PCA and ADMIXTURE) are not reliable for detecting ancient admixture events (several 100s or 1000s of generations back in time). Despite the name, ADMIXTURE shows population clustering or structure rather than a clear signature of gene flow. The examples here are also likely the result of recent hybridization, as suggested by the authors. Hence, I suggest renaming this paragraph to "Population clustering", rather than "Inferring admixture", because the term "admixture" is often used somewhat differently in population genetics. This may be a minor task, but relevant.

4) Another minor point: The gene lists are interesting, and the overlap with other studies is certainly of value. Considering that there is no enrichment for GO terms, it might be worthwhile testing for formal enrichment in KEGG pathways as well.

Specifically, we ask that you address the concerns raised by Reviewer #1 regarding the possibility for the observed signatures to be caused by natural selection rather than domestication. We also ask that you address the comments of Reviewer #2 asking for additional genomes, and that you use the same selection tests for all branches, as requested by Reviewer #3.

If the revision process takes significantly longer than six months, we will be happy to reconsider your paper at a later date, as long as nothing similar has been accepted for publication at Communications Biology or published elsewhere in the meantime.

We are committed to providing a fair and constructive peer-review process. Do not hesitate to contact us if you wish to discuss the revision or if there are specific requests from the reviewers that you believe are technically impossible or unlikely to yield a meaningful outcome.

Reviewers' comments:

Reviewer #1 (Remarks to the Author):

The authors analyze genetic diversity within and among three camel species to infer aspects of their population history and to identify genes which may have been targeted by selection, particularly during the domestication of two of the species. The results are presented in the context of the domestication syndrome and various proposed hypothesis in terms of gene pathways whose alteration lead to domestic phenotypes. This makes the findings likely to be of interest to a broad audience. As such, the manuscript would benefit from inclusion of additional relevant details about the history and divergence of the studied species. For example, the divergence between bactrianus and ferus (~1 MYA) should be made clear to highlight for the reader that ferus is not a proxy for the 'pre-domesticated ancestor' of bactrianus and to aiding in the interpretation of the used selection statistics. Similarly, the phylogenic relationship to dromedaries at ~5-8 MYA should be emphasized in the text to aid the reader. If I understand correctly, this makes dromedaries a separate domestication from a genetically quite distinct and diverged ancestor vs the two-hump camels. This greatly complicates the comparisons among the two domesticated species. Because of these differences, the tests for selection, and the signals and time scales that might be detected, are quite different.

Response: The reviewer is correct; the two domesticated camel species arise from two independent domestication events from different wild ancestors – which is essentially a strength of our comparison that we have already highlighted in the discussion [lines 338 – 348]. Nonetheless, we have added this information, including a description of the phylogenetic relationship among the three species and a statement regarding the extinction of the wild ancestor of each, to the introduction to better prepare the reader for interpreting the selection results [lines 123 – 127]. Please see our further responses below for clarification of why different tests of positive selection were chosen for each scenario.

Further, because of the time scales, it is a concern that the tests may identify effects of natural selection along the lineages (millions of years) instead of recent effects associated with domestication. The logic excluding this possible confounding should more clearly be described.

Response: The tests of selection chosen are specific for the relevant taxa and assumptions of the tests, and are consistent with more recent events rather than more ancient events. Methods related to Ka/Ks ratios and gene family expansion/contraction are on the scale of millions of years, and are thus more indicative of camel-specific adaptations rather than domestication, and have been discussed in previous single-genome studies (Jirimutu et al. 2012, Wu et al. 2014). Except for the PBS test (which focuses on very recent events due to rapid allele frequency differentiation), we used tests based upon comparisons of polymorphism to divergence (either within genes or across the entire genome) that identify moderate to recent selection events. We do agree with the reviewer that we cannot completely exclude the possibility of positive selection acting on the wild ancestor of the domesticated species (thus not a result of domestication), and have indicated this in the discussion [lines 343 – 348] for clarification.

Since there are multiple genomes available, the authors should consider using MSMC (Schiffels 2014) to both obtain population size profiles for more recent times and to better define the dynamics of population separation between the species.

Response: We appreciate the reviewer's suggestion on the use of MSMC. We had previously considered the use of this method, but for the reasons outlined below refrained from its use. MSMC explicitly requires accurately phased haplotypes as the input (Schiffels and Durbin 2014). Because the *Camelus ferus* reference genome remains in a draft form of >13000 scaffolds, and our largest within-population sample size is 9, generating accurately phased genome-wide haplotypes is challenging. Schiffels and Durbin (2014) did test the MSMC method on unphased genotypes as well, but reported that "biases occurred at the two ends of the analyzed time range..." and thus "left unphased sites in the analyses of population size estimates but removed them from the analyses of the population splits". Therefore, we have elected to keep the PSMC results in the manuscript, as this method is still considered a standard in the field and robust to a more fragmented reference genome and unphased genotypes (as compared with humans, for example). Furthermore, as we have discussed in lines 209 – 215, we already acknowledged that our interpretation could be affected by known population splits and migration, but demonstrate consistency in our results with those from previous studies.

I believe that alpaca is also considered to be domesticated, the author's may which to comment on what impact the use of a domesticated samples as an outgroup may have.

Response: Yes, the alpaca is a domesticated species, but no phylogenetic analyses were performed using alpaca's as an outgroup. The cladogram shown in Figure 1 uses alpacas to root the tree, but this is simply for visual purposes as no phylogenetic analysis was performed. Next, the only analyses we performed that required the explicit use of an outgroup were the PBS

tests, where dromedaries form the outgroup by nature of the three-taxon tree among Old-World camelids. Finally, we did use the alpaca genome in the homogeneity/HKA tests for determining selection in dromedaries. Under neutrality, this test assumes that within and between species polymorphisms are correlated. Thus, dromedary polymorphism was compared to divergence between both wild camels and alpacas as more extensive support of effects specific in dromedaries. As a result, selected genes in dromedaries are largely divergent from both species. It could be argued that if the exact same SNP allele was positively selected in both species it could reduce the ability to detect positive selection in dromedaries. First, this generally supports our conclusions of positive selection specifically in dromedaries. Second, this would be extremely unlikely, and the nature of the test requires multiple polymorphic sites within each gene thus rendering this possibility even more unlikely and thus of little effect on our conclusions.

Reviewer #2 (Remarks to the Author):

The presented data do not fully support the main conclusions of camel domestication because they created a big gap in sampling, by the way the paper has worthwhile information that is useful for the signature of selection in domestication specially in old world camels , and there are essential points missing in the manuscript which need to be addressed in detail.

Line 137: The sequencing depth is low, while as reported the WGS data with a medium degree of depth setting (more than 15x) can be considered as one of the recommendation for SNV calling," Kishikawa, Toshihiro, et al. "Empirical evaluation of variant calling accuracy using ultra-deep whole-genome sequencing data." Scientific reports 9.1 (2019): 1784."

Response: We thank the reviewer for providing this recent citation benchmarking SNP calling at various depths. It is unclear why the reviewer suggested our sequencing depth is low, whereas it appears markedly consistent with recommendations by the referenced study (Kishikawa et al. 2019). Kishikawa et al. suggested ~15X depth for an ideal balance of genotype concordance and costs. Our uniquely mapped read depth (URD in Kishikawa et al.) ranged between 12.5-14.6X. Although ever-so-slightly less than the 15X recommendation, Kishikawa et al. reported that 13.7X depth resulted in >99% genotype concordance. Thus, our URD can be considered as producing highly accurate genotype calls. Furthermore, we included arguably more stringent criteria for including SNPs than did Kishikawa et al. (e.g., the overlap of multiple SNP callers and additional filters), also supporting the accuracy of our genotypes. Finally, it is worth mentioning that the combination of practices reported by Kishikawa et al. to produce the highest genotype concordance rates were identical to those used in our study: GATK best practices + HaplotypeCaller + VQSR. In light of this new reference, we have added it to further support the accuracy of our genotype data [lines 166 – 169, ref #32]

Line 137: the authors should explain more about the reason for using camelus ferus as a reference genome in alignment.

Response: The *C. ferus* reference genome was the most complete and contiguous camelid genome available at the time (~13k scaffolds compared with >30k for the other species). Furthermore, at least for dromedaries which are the most distantly related of the three species, we repeated the entire analysis by mapping all sequences to the dromedary reference genome (intraspecific mapping) and produced nearly identical results [lines 176 – 178; Table 1; Fig 2]. Next, since the ultimate goal was to identify SNPs within and across species at homologous sites, using only 1 reference genome was optimal, and coverage within each species was high enough and similar enough across species that interspecific SNP-calling biases can be considered non-existent (again, see Table 1 & Fig 2). Finally, we indicated that previous studies [lines 157 – 163], including whole genome comparisons performed by out group and others, have shown substantial amounts of genome synteny existing across camelids thus supporting the use of a single reference genome.

Line 163: the ratio ts/tv appears to be high (2.54) that usually it happens in exome studies or small genomes with high GC for example turkey genomes, the authors should explain about this.

Response: The ts/tv ratio has generally been used as a standard criterion for quality of SNP calling. At random, the expected ts/tv ratio is 0.5, but in reality, with known variants in humans, this ratio is ~2.1. Ratios much less than 2 suggest many false positives, and extremely high ratios $\gg 3$ could be indicative of possible biases (see DePristo et al. 2011, Liu et al 2012, Baes et al. 2014). Human exome data is often in the 3 – 3.5 range (see DePristo et al. 2011). As a result, our ts/tv ratio is 1) well above any indication of numerous false positives, 2) well below that of extreme cases such as exome data, and 3) consistent with other species, such as turkey, whose GC content (41.6%) is similar to camels (41.5%). Both turkeys and camels have higher GC content than humans (40.9%), which is consistent with a higher ts/tv ratio. Furthermore, the ts/tv ratio can also vary depending upon regional GC content, methylation levels, mutation rates, etc. Thus, the true ts/tv ratio is difficult to know. Nevertheless, our results remain consistent with that of a high-quality SNP dataset and lacking obvious biases. At this time, we will refrain from additional mention of this in the manuscript in order to save space for addressing more critical concerns.

Line 165: the author would be defining fixed differences (SNP or variants) to clarify for public readers.

Response: We have added the parenthetical “monomorphic within a species, but differed between species” to help clarify this distinction [lines 171 – 172].

Line 165-170: when you used the *C. ferus* as reference genome, we expected the numbers of variants in the dromedary vs *C. ferus* set (2722670SNP) should be more than Bactrianus vs *C. ferus* (5188560SNP) and *C. ferus* vs *C. ferus* sets (3912111 SNP), but it was vice versa. The author must explain why it happens. In addition recheck the number of total variant (10.8 million) that seems to be 11.8.

Response: We appreciate the reviewer's concern in checking the SNP counts. We checked our files again, and identified 10.8 million SNPs as reported [lines 169 – 172]. ~2.1 million SNPs were fixed between species, and thus not shown in Fig 1d as correctly stated in the caption [lines 834 – 842]. The remaining number of segregating, polymorphic SNPs reported in Fig 1d total 8.7 million ($8,709,216 = 2045841 + 520658 + 107662 + 48509 + 2230606 + 2329634 + 1426306$). Unfortunately, the reviewer accidentally failed to account for the overlap of SNPs between species in their math and thus inflated that estimate by approximately a million. 10.8 million SNPs is correct.

Second, the reviewer appears concerned that overall, we found more SNPs in Bactrians (5.2 million) and wild camels (3.9 million) compared with Dromedaries (2.7 million). It is unclear why the reviewer believes the opposite should be true, as these numbers mirror the overall levels of genetic variation known from previous camelid genomes (Jirimutu et al 2012; Wu et al. 2014; Burger and Palmieri 2014; Fitak et al. 2016). For example (excluding our results in this study), genome-wide heterozygosity in *C. bactrianus* is $1 - 1.3 \times 10^{-3}$ (Jirimutu et al 2012; Wu et al. 2014; Burger and Palmieri 2014), and *C. ferus* is similar although bit higher (Jirimutu et al. 2012). However, dromedaries have a markedly lower (25-35% less) amount of genetic variation between $0.71 - 0.74 \times 10^{-3}$ (Wu et al. 2014; Fitak et al. 2016). Thus, as we have already discussed in the manuscript [lines 188 – 189], our findings are noticeably consistent with every camel genomic study to date: dromedaries have substantially less variation than other camelids. If for any reason the reviewer is concerned that this could be a result of the interspecific read alignment, the entire analysis within dromedaries was repeated with the intraspecific reference with essentially identical findings (2,818,163 SNPs, see Table 1, lines 147 – 157; 188 – 189; 475 – 478). A more detailed discussion of the number of SNPs within and between species can be added if the Editor deems it necessary, but at this time we include the appropriate results and references, and focus the text on the description of more critical findings.

Also, it would be worthwhile to add the result of InDels and compare between species.

Response: Of course INDELS, and more generally structural variants, are important functional aspects of genomes and as the reviewer suggests, the use of INDEL polymorphisms may improve the resolution of our inferences by increasing the number of markers. However, INDELS have generally been excluded in population genomic analyses, especially of non-model species, for a variety of reasons. First of all, their accurate genotyping is still quite challenging as little INDEL detection concordance exists among programs – owing to a suite of factors related to coverage, read alignment, library preparation, sequence context such as homopolymers, etc. (see reviews by new ref #76). It has been estimated that much higher coverage ($\geq 60X$ for Illumina data) (new ref #77) is needed for accurate INDEL detection. Furthermore, draft genomes of non-model species often contain numerous collapsed regions, notably among repetitive elements, which hinders the ability to accurately genotype INDELS. Then, because they are orders of magnitude less frequent than SNPs, they contribute very little to the statistical analyses detecting selection. Finally, if an INDEL is the actual variant under positive selection, the results will still be evident in patterns of polymorphism at the linked SNP loci and thus easily detected (the window is thus detected, but not the actual causative variant)

by our analyses. Again, we are not suggesting that INDEL polymorphisms are not important genetic markers, but rather SNPs are currently the most robust marker for the design of the current study. We have added a citation and a brief discussion of this in the manuscript [lines 346 – 350] to encourage future research in this area using methodology specific to INDEL detection:

“Future studies targeting the analysis of insertion/deletion (indel) polymorphisms may be useful for identifying additional targets of selection⁷⁶. Indel polymorphisms were omitted from our analyses because their accurate genotyping remains quite challenging, especially without high coverage ($\geq 60\times$ ⁷⁷) and in non-model species with incomplete, or draft genome assemblies⁷⁸”

Line 355: the distribution of sampling for decisions about genomic signature of domestication is not adequate because the author create long gaps of the world sampling (north Africa, IRAN, and other central Asia for dromedary and china, Russia, Iran and etc. for Bactrian) that are important gene pool for camels, in my suggestion if it is difficult to sampling of these aria, the authors could get the deposited data through NCBI.

Response: Please see our response below to reviewer #3 who presented a similar comment.

Line 392: what is the reason of using the cattle as reference for alignment for X and Y chromosome, the author would be explaining about this.

Response: We have added the statement [lines 412 – 41f] to help clarify this: “This was a necessary step in order to remove variants from downstream analyses that require accurate estimates of allele frequencies assuming diploid samples (our samples consisted of both the homogametic and heterogametic sexes).” Essentially, the cattle genome is the most closely related species with a complete, chromosome-level genome assembly. Thus, it was the optimal choice for anchoring scaffolds to putative sex chromosomes. There is also a high degree of karyotypic synteny known between camels and cattle (see our reference to Balmus et al. 2007; lines 157 – 160). Finally, our use of the cattle sex chromosomes as a reference is consistent with that reported in the study by Wu et al. 2014, where LASTZ was also employed, with nearly identical parameters, for aligning camel scaffolds to the cattle reference genome (notably for the X chromosome). This renders our results more comparable with previously published work. We refer the reviewer to the complete description of the methods [lines 411 – 426] and also to the GitHub code repository (https://github.com/rfitak/Camel_Genomics) for bioinformatic details of how the work was performed.

In the end of comments, the authors compare the positive selected genes with other species such as fox, dogs but it would be useful to compare with camels that reported by Wu et al., 2014.

Response: We had initially decided to only compare genes under selection with those from other systems (species) related to domestication, since the positively selected genes reported

by Wu et al. target a deeper evolutionary divergence and are thus more related to general camel adaptation rather than domestication. Nonetheless, at the reviewer's request we have added a description of the overlapping positively selected genes with camels reported in Wu et al. [lines 256 – 262]. Only positively selected genes from Dromedary overlapped (n=4), as no overlap in domestic Bactrian camel genes existed. The added text states:

“Four genes were previously identified as under positive selection in both *C. dromedarius* and *C. bactrianus* (CENPF, CYSLTR2, HIVEP1) or just in *C. dromedarius* (CCDC40) (Wu et al. 2014), suggesting either a convergent consequence of domestication, or a more general role in camel adaptation. The latter is more likely, considering that CENPF, CYSLTR2, and CCDC40 are each related to ciliopathies and/or respiratory diseases such as asthma – indicating a possible adaptation to the respiratory challenge posed by dust in highly arid environments (Wu et al. 2014).”

Reviewer #3 (Remarks to the Author):

The paper is very well-written, and provides novel insights into the domestication of camels. This has implications for genetic pathways involved in domestication in other species as well, which is certainly important for people in the field. The methods seem appropriate and technically sound, and are very well-documented. The results are interpreted appropriately, and context is provided for a general understanding of each part. In principle, only few things should be considered:

Response: We appreciate the positive remarks by this reviewer.

1) Major point: A study on 128 whole genome sequences of Asian camels was published on bioRxiv recently (doi:10.1101/656231v1). It would be good to discuss their findings regarding demographic history and admixture (methods like PCA or PSMC were also applied there), since there is an overlap in scope and results. This does not imply the results were identical or mutually exclusive, but seem rather complementary, and it would be important to be discussed here. Despite being available only on bioRxiv, it is common in population genetics to consider such manuscripts, and I recommend doing so.

Response: In principle, we agree with the reviewer to include published data, however in this case we have major concerns. 1) During the creation of the data and beginning of the analysis for this manuscript, there were no other camel genomic data available apart from those we created and present in this manuscript. The data available in NCBI as of the end of 2019 and there was no chance to timely include them in our analysis. 2) - and more severe, we have major ethical and scientific concerns about the sequences published in bioRxiv recently (doi:10.1101/656231v1): The sampling of the wild Bactrian camels has not been approved by the wild camel protection foundation (WCPF), which is by legal agreement with the Mongolian Ministry of Agriculture and Tourism the only organization responsible for the semi-captive wild camel breeding herd, from which the samples were taken. The corresponding author (Pamela Burger) has respective correspondence with the WCPF, which is confidential, but if absolutely necessary can be shown to the handling editor upon request. 3) We have qualified evidence

that camel samples (mainly dromedaries) from Iran and Central Asia used in the bioRxiv study are introgressed with other taxa. As our manuscript deals with the evolutionary and domestication history of the separate species, those (highly likely) introgressed samples would not add useful information to our analyses.

We therefore prefer to use only our own sequence data, which we absolutely know where they come from and how they were sequenced, for this manuscript – even more considering that there is little genetic differentiation and population structure in camels in general, which has been shown multiple times (Almathen et al. 2016, Chuluunbat et al. 2014, Li et al. 2017) [lines 377 - 383].

Notwithstanding, we are already in the preparation phase of a collaborative proposal together with the first author of Ming et al. (2020) to expand and continue the analysis on a joint data set.

2) Another major point: I cannot follow the reason for applying three different selection tests on the three branches. It would make sense to at least provide the results for all types of tests for all three branches as Supplementary, and provide a rationale for focusing on specific tests for each population. At least for *C. bactrianus* and *C. ferus*, there should be the same power for the same tests, for example using the PBS statistic.

Response: We appreciate the reviewer's concern regarding the choice of test for each of the three species. The tests chosen were determined after careful consideration of the assumptions for each, and could not be performed in each species. Ultimately, our tests were focused on identifying positive selection related to domestication, and thus we have changed the title of the section '*Selection in wild two-humped camels*' to '*Relaxed selection relative to wild two-humped camels*'. We have also ensured that in the manuscript we have clarified we are not detecting positive selection specifically in wild camels. As for the specific choice of tests:

- 1) As requested by the reviewer, we have added the results of the homogeneity/HKA test result in Bactrian camels to the text (only 90 genes could be tested, 0 under positive selection; lines 288 – 292). We apologize for not including this table in the initial draft.
- 2) The PBS test can only be performed in either domestic Bactrian or wild camels. The PBS test assumes a pair of sister taxa, and a third outgroup taxa. As a result, this test could only be performed in dromedaries if we had population genomic data from another more distantly related outgroup (which doesn't exist for the Camelini). Within wild camels, we did not report the PBS test results because a) we are focused strictly on selection related to domestication, and b) comparing a wild species to two different domestic species would artificially inflate the overall PBS values, leading to numerous false positives (as opposed to comparison just within the domestic Bactrian camel).
- 3) The test for relaxed selection in domesticated species theoretically includes windows possibly under positive selection in wild camels. The rationale for this decision is that regions of low polymorphism and negative Tajima's D in wild camels are indicative of positive and/or background selection, and when combined with overlapping regions of extreme excess of polymorphism in both domesticated species, indicates a lack of

directional selection in this region. We use these windows to suggest genes that have experienced the relaxation of selection as a result of domestication.

In summary, the homogeneity/HKAA test combination was performed for both domestic species, looking at windows of low polymorphism/high divergence was performed in both domestic species, and the PBS test was only performed in Bactrian camels where it's assumptions were valid. In wild camels, we did not perform tests for selection, but rather used them for detecting relaxed selection using #3 above. This does not imply a lack of importance of detecting positive selection in wild camels, but rather it is beyond the scope of our study related to domestication, we currently lack the correct interspecific comparisons, and the extremely small population size of wild camels renders differentiating positive selection from demographic effects challenging.

3) It should be noted that the applied methods (PCA and ADMIXTURE) are not reliable for detecting ancient admixture events (several 100s or 1000s of generations back in time). Despite the name, ADMIXTURE shows population clustering or structure rather than a clear signature of gene flow. The examples here are also likely the result of recent hybridization, as suggested by the authors. Hence, I suggest renaming this paragraph to "Population clustering", rather than "Inferring admixture", because the term "admixture" is often used somewhat differently in population genetics. This may be a minor task, but relevant.

Response: We appreciate this clarification and have made the requested change [lines 217; 452].

4) Another minor point: The gene lists are interesting, and the overlap with other studies is certainly of value. Considering that there is no enrichment for GO terms, it might be worthwhile testing for formal enrichment in KEGG pathways as well.

Response: We appreciate this suggestion and have added the analysis to the results [lines 246; 298] and Methods [lines 546 – 551]. Again, no significant enrichment was found.

Reviewers' comments:

Reviewer #1 (Remarks to the Author):

The changes made by the authors have improved the clarity of the presentation and the revised manuscript has addressed the major concerns.

Reviewer #3 (Remarks to the Author):

As in the first round of reviews:

1) It remains curious that the ethical issue on another manuscript/dataset raised by the authors of this paper did not prevent a final publication of that manuscript as a paper in the same journal, *Communications Biology* (<https://doi.org/10.1038/s42003-019-0734-6>). It is even more curious that the authors declare to start a collaboration with the authors of that study – if these samples are ethically so problematic that the data should not be used and their findings not even mentioned, how can they be used in a future collaboration?

Further, the authors argue that a comparison is not reasonable due to introgression in individuals from the other dataset. Meanwhile, some individuals in their study are also claimed to show introgression – 6% is not a “small” amount when compared to admixture studies, for example, in humans.

Apart from that, there was obviously a misunderstanding in the point I was trying to make: I did not suggest to include the published data to the analysis of this study, and I agree that given the speed at which genomes are sequenced one cannot be at all times fully inclusive. I merely suggested to discuss (!) the findings concerning population history and the relationships of the different species (compare Fig. 1 this study and Fig. 2 in Ming et al.). This simply means comparing the state of knowledge on the research topic, instead of not at all mentioning the only other comparable dataset for this clade. This could even include a statement within the manuscript why the authors believe their data and conclusions are more consistent than others. I believe it is common practice to discuss findings of others that were published before publishing one's own study, but after generating one's own data, since it takes long time between data generation and publication. I understand that the authors are reluctant to apply modifications to the manuscript, but elaborating in the responses to the reviewers might not help the reader.

In my opinion, this is important, but I have nothing at stakes here personally. Hence, the editor needs to decide whether the ethical concern justifies ignoring results from the only study of similar scope and published in the same journal, or not.

2) This is well explained now. However, I would amend that the whole “positive selection” section discusses candidate genes, putatively selected genes, regions potentially under selection.

In principle, reviewers could ask for a power analysis: given the demography, what is the expectation for the p-value or Q distribution? Can simulations be used to obtain more reliable values? The evidence for positive selection in this manuscript is weak, with a main point of the manuscript being putative implications of it. To be clear: I don't request such simulations, but I think it should be more clear in the main text (not only the methods section) that “candidate genes”, “putatively selected genes” are presented, at each and any instance where these genes or gene sets are addressed. Additionally, I do think that the use of the Q score is somewhat confusing, since here this cutoff represents nothing else than a nominal p-value cutoff of 0.05. The point of this conversion in Liu et al. 2014 was that it provides a ranking that can be used in enrichment tests (for example, Wilcoxon ranked test, or the GOrilla ranked test applied in Liu et al.). Since here only a classic 2x2 test for enrichment was applied, this does not make much sense.

To be clear: a) I see no sense in the Q scoring. b) The phrasing regarding evidence for selection must be addressed throughout the manuscript, including the abstract.

3) Having another look at the methods paragraph: Again, not "admixture" is inferred here, but "ancestry".

4) -

Additionally, I suggest that it is pointed out that the number of SNPs in *C. bactrianus* (Fig. 1d) is based on 7 individuals compared to 9 for the other branches. Hence, for the same number of individuals, the relative number of polymorphic sites will be even higher in *C. bactrianus*. Even though the authors reject discussing such aspects of the data (in response to reviewer #2), it is worthwhile. A valuable representation (and independent from population-wise estimates from different numbers of individuals) would be a table providing the numbers of heterozygous sites and their proportion of all confidently called sites (i.e. simple heterozygosity) for each individual separately (and a boxplot from this). This is a very standard representation of genomic diversity in other mammalian clades, and will be a very useful support for understanding the differences between the populations.

Reviewers' comments:

Reviewer #1 (Remarks to the Author):

The changes made by the authors have improved the clarity of the presentation and the revised manuscript has addressed the major concerns.

Response: We appreciate the positive endorsement by this reviewer.

Reviewer #3 (Remarks to the Author):

As in the first round of reviews:

1) It remains curious that the ethical issue on another manuscript/dataset raised by the authors of this paper did not prevent a final publication of that manuscript as a paper in the same journal, Communications Biology (<https://doi.org/10.1038/s42003-019-0734-6>). It is even more curious that the authors declare to start a collaboration with the authors of that study – if these samples are ethically so problematic that the data should not be used and their findings not even mentioned, how can they be used in a future collaboration?

Further, the authors argue that a comparison is not reasonable due to introgression in individuals from the other dataset. Meanwhile, some individuals in their study are also claimed to show introgression – 6% is not a “small” amount when compared to admixture studies, for example, in humans.

Apart from that, there was obviously a misunderstanding in the point I was trying to make: I did not suggest to include the published data to the analysis of this study, and I agree that given the speed at which genomes are sequenced one cannot be at all times fully inclusive. I merely suggested to discuss (!) the findings concerning population history and the relationships of the different species (compare Fig. 1 this study and Fig. 2 in Ming et al.). This simply means comparing the state of knowledge on the research topic, instead of not at all mentioning the only other comparable dataset for this clade. This could even include a statement within the manuscript why the authors believe their data and conclusions are more consistent than others. I believe it is common practice to discuss findings of others that were published before publishing one's own study, but after generating one's own data, since it takes long time between data generation and publication. I understand that the authors are reluctant to apply modifications to the manuscript, but elaborating in the responses to the reviewers might not help the reader. In my opinion, this is important, but I have nothing at stakes here personally. Hence, the editor needs to decide whether the ethical concern justifies ignoring results from the only study of similar scope and published in the same journal, or not.

Response: We understand the reviewer' concern and we do not disagree with any of the findings by Ming et al., but rather with the means of collecting their 19 wild camel specimens. The camel research community is small, and this disagreement has not inhibited the establishment of collaborations between research groups. None of the questionably collected

19 specimens are part of the potential collaboration, [REDACTED] as the ultimate goal of characterizing and conserving domestic and wild camel genomic resources is paramount. Of course, through this collaboration, any potential future specimen collections will be made through the appropriate channels.

With regards to the remaining comment, we have included additional comparisons of our results with Ming et al. (lines 180 – 182; 194 – 199; 241 – 243). Many of these comparisons show the strong congruence between our studies, especially as it relates to measures of variation in *C. bactrianus* and *C. ferus*, and the marked similarity in inferred admixture proportions. We also highlight the one major difference, where Ming et al. found 3x higher variation in dromedaries. According to Ming et al., this is likely the result of highly admixed dromedaries and the small sample size, as it contradicts previous findings by both our group (this study and others) and the Jirimutu group (lead author of Ming et al.)

Lines 180-2: “The large number of SNPs shared between the domesticated species is consistent with known introgression events and has been observed in other genomic studies of camels³⁶.”

Lines 194-199: “Interestingly, Ming et al.³⁶ reported similar patterns of genetic variation in *C. bactrianus* ($\pi = 0.95 \times 10^{-3} - 1.1 \times 10^{-3}$) and *C. ferus* ($\pi = 0.88 \times 10^{-3}$ compared with 0.71×10^{-3} in this study), but nearly 3-fold higher levels in *C. dromedarius* ($\pi = 1.5 \times 10^{-3}$). Although this finding by Ming et al. conflicts with our results and that of the previously mentioned studies^{20,22-24}, the authors attributed this to the small sample size ($n = 4$ dromedaries, but at least one was removed for genetic similarity) and being sampled from Iran where hybridization with *C. bactrianus* is commonplace^{36,38}.”

Lines 241-243: “These values are markedly similar to values reported by Ming et al.³⁶, where *C. dromedarius* ancestry in *C. bactrianus* ranged between 1 – 10%, and *C. bactrianus* ancestry in three *C. ferus* individuals ranged between 7 – 15%.”

2) This is well explained now. However, I would amend that the whole “positive selection” section discusses candidate genes, putatively selected genes, regions potentially under selection.

In principle, reviewers could ask for a power analysis: given the demography, what is the expectation for the p-value or Q distribution? Can simulations be used to obtain more reliable values? The evidence for positive selection in this manuscript is weak, with a main point of the manuscript being putative implications of it. To be clear: I don’t request such simulations, but I think it should be more clear in the main text (not only the methods section) that “candidate genes”, “putatively selected genes” are presented, at each and any instance where these genes or gene sets are addressed. Additionally, I do think that the use of the Q score is somewhat confusing, since here this cutoff represents nothing else than a nominal p-value cutoff of 0.05. The point of this conversion in Liu et al. 2014 was that it provides a ranking that can be used in enrichment tests (for example, Wilcoxon ranked test, or the GOrilla ranked test applied in Liu et al.). Since here only a classic 2x2 test for enrichment was applied, this does not make much sense.

To be clear: a) I see no sense in the Q scoring. b) The phrasing regarding evidence for selection

must be addressed throughout the manuscript, including the abstract.

Response: a) We agree with the reviewer that assessing statistical power with regards to determining positive selection is challenging. As a result, we have removed the conversion to Q score, and rephrased the section to be more consistent with the intent of a ranked order of prioritization rather than statistical significance (lines 518-524; Supplementary table 3). We have also indicated throughout the manuscript that these are “putative|candidate” positively selected genes (e.g., Abstract Line 43; Results Lines 250, 258, 287, 310; Discussion Line 345; Methods Lines 496, 518, 526, 530, 564).

Lines 518-524: “For the final set of putative positively selected genes, we retained those with a homogeneity test $P < 0.05$ and with a significant HKA test score ($P < 0.05$) only in the dromedary population. As suggested by Liu et al.⁴⁹, the P-values obtained from these tests can be misinterpreted since accurate P-values can be only be obtained from simulations, but can be informative when combined with other ranking criteria. In conjunction with the recommendation by Liu et al., we emphasize that these genes are in ranked order of priority, or evidence, rather than of statistically significant effect.”

3) Having another look at the methods paragraph: Again, not “admixture” is inferred here, but “ancestry”.

Response: We have corrected this in the methods, and also Fig 1, by changing ‘admixture’ to ‘population clustering’ (lines 468, 865).

4) Additionally, I suggest that it is pointed out that the number of SNPs in *C. bactrianus* (Fig. 1d) is based on 7 individuals compared to 9 for the other branches. Hence, for the same number of individuals, the relative number of polymorphic sites will be even higher in *C. bactrianus*. Even though the authors reject discussing such aspects of the data (in response to reviewer #2), it is worthwhile. A valuable representation (and independent from population-wise estimates from different numbers of individuals) would be a table providing the numbers of heterozygous sites and their proportion of all confidently called sites (i.e. simple heterozygosity) for each individual separately (and a boxplot from this). This is a very standard representation of genomic diversity in other mammalian clades, and will be a very useful support for understanding the differences between the populations.

Response: We have clarified both in the text (lines 172 – 175) and in the caption for Fig 1d (lines 871 – 872) that only 7 individuals were included for *C. bactrianus* as opposed to 9 individuals for the other species. We agree with the reviewer that a plot of individual heterozygosity would be a useful comparison, and we have updated Fig 2 with a panel of this (part a). Rather than an additional table, the raw data for this plot, as well as for the entire study, have been deposited into Dryad with the tentative link provided in the ‘Data Availability’ section.

Lines 172 – 175: “The most segregating SNPs were observed in *C. bactrianus* (5.2 million), much higher than observed in *C. ferus* (3.9 million) and *C. dromedarius* (2.7 million)

and despite sampling fewer individuals ($n = 7$ for *C. bactrianus* compared to $n = 9$ for the other two species).

Lines 871 – 872: “Note that the number of *C. bactrianus* individuals ($n = 7$) was less than the number of *C. ferus* and *C. dromedarius* individuals ($n = 9$).”

Fig 2a:

REVIEWERS' COMMENTS:

Reviewer #3 (Remarks to the Author):

I thank the authors for their understanding concerning the issues I raised. I think that my points are very well addressed, and the manuscript has been improved. Additionally, more materials have been made public, which I appreciate very much.